# Program evaluation of a school-based mental health and wellness curriculum featuring yoga and mindfulness

**Bethany H. McCurdy**[1]*, **Travis Bradley**[2], **Ryan Matlow**[2], **John P. Rettger**[2], **Flint M. Espil**[2], **Carl F. Weems**[1], **Victor G. Carrion**[2]

1 Human Development and Family Studies, Iowa State University, Ames, Iowa, United States of America,
2 Department of Psychiatry and Behavioral Sciences, School of Medicine, Stanford University, Stanford, California, United States of America

* bmccurdy@iastate.edu

**Data Availability Statement:** All data files are available via GitHub (https://github.com/bmccurdyISU/HealthandWellnessStudy).

## Abstract

### Background

Interest in the effectiveness of mindfulness-based interventions such as yoga in primary schools has grown. Evidence shows promise, as youth who engage in yoga to promote mindfulness show improved coping skills, increased socio-emotional competence and pro-social skills, academic performance, attention span, and ability to deal with stress.

### Objective

This study reports the results of a program evaluation of a universal health and wellness curriculum, Pure Power, designed to teach youth yoga techniques, mindfulness, and emotion regulation.

### Methods

A non-randomized comparison design examined outcomes among participants from schools that completed the intervention with highest fidelity of implementation (n = 461) and from students in matched comparison schools (n = 420). Standard measures of coping, emotion regulation and emotion dysregulation, spelling, and math achievement were collected.

### Results

Analyses suggest the youth in the intervention schools demonstrated relative improvement on measures of emotion regulation, spelling, and math.

### Conclusions

Challenges in implementation in real-life settings are vital to identify. The data provide some real-world evidence for the effectiveness of a universal health and wellness curriculum on emotion regulation and positive academic outcomes. Training school staff to deliver the intervention may foster implementation. Future research should test the effectiveness of

**Funding:** This project was supported by the Lucile Packard Foundation for Children's Health and NIH award UL1 TR001085. The funders had no role in study design, data collection and analysis, decision to publish, or preparation of the manuscript.

**Competing interests:** The authors have declared that no competing interests exist.

who delivers the intervention; for example, teacher-delivered groups vs. other wellness personnel.

## Introduction

Children who experience traumatic and adverse life experiences (TRACEs) [1] are at an elevated risk of behavioral and mental health concerns [2, 3]. TRACEs involve experiencing a threat (e.g., physical, sexual, and emotional abuse, exposure to violence) or deprivation (e.g., physical and emotional neglect, parental separation) and the types of adversity, duration of adversity, cumulative experiences, and interactions among experiences of adversity are all important in a child's development [4]. Duration of stressors may be acute or chronic, such as whether a child experienced an environmental disaster or was born into poverty [4]. Children's perception and interpretation of adversity are associated with a myriad of challenges in emotional development, and following adversity, children may experience further social and academic challenges [5, 6]. School-based interventions that target general wellness may foster resilience in the wake of adversity, and so are an important avenue for primary prevention of mental health problems and to improve academic achievement among youth at risk for TRACEs [7].

In the face of stress and adversity, children who develop adaptive coping and emotion regulation strategies are less likely to experience negative effects and be better at managing stressful situations [8, 9]. An integrative model of coping strategies includes three broad strategies that have received attention in empirical studies [10]. First, children may engage in the *solution-focused* strategy, which includes identifying and seeking solutions to resolve problems. Second, the *emotion-oriented* coping strategy focuses on emotion regulation or the ability to regulate negative emotional states, such as anxiety, shame, fear, guilt, anger, or despair. Third, children may engage in an *avoidant-oriented* strategy, which entails avoiding stressors or adversity by distancing in various ways [11].

Mindfulness and mindfulness training have been shown to have positive effects on coping and emotion regulation. Mindfulness is described as "paying attention in a particular way: on purpose, in the present moment, and nonjudgmentally" [12]. Mindfulness encourages non-judgmental attention to the present moment while reducing negative feelings stemming from unpleasant experiences [13]. In children, interventions utilizing mindfulness may support emotion regulation and healthy coping mechanisms in response to stress [14]. Effectively, mindfulness is thought to enhance the ability to engage in solution-focused strategies and emotion regulation through promoting goal orientation, problem disengagement, and mobilizing resources to focus on acceptance and awareness while minimizing avoidance-oriented coping by fostering an attitude of non-attachment towards stressful life experiences [15, 16].

A notable and widely used set of techniques that can promote mindfulness is yoga [17]. Yoga is a mindfulness-based practice that combines breathing, posture, meditation, and movement [18]. Decades of research on adults find that yoga and mindfulness instruction aid in addressing the negative impacts of stress and adversity on emotional and behavioral functioning by reducing feelings associated with anxiety and depression while improving attention modulation [14]. In adult populations with health concerns, a meta-analysis found that yoga and mindfulness programs effectively manage and reduce stress [19]. Interest in the effectiveness of mindfulness-based interventions such as yoga in primary schools has grown. Preliminary evidence shows some promise, as youth in second grade to twelfth grade who engage in yoga to promote mindfulness showed improved coping skills, increased socio-emotional

competence and prosocial skills, decreased alcohol use, and teacher-rated improvements in academic performance, attention span, and ability to deal with stress and feelings of anxiety [20–24].

Overall results on mindfulness-based interventions in youth are mixed [22, 23]. For example, a multi-year longitudinal study [25] measuring the effectiveness of a mindfulness-based stress reduction intervention in 12–18-year-olds found that youth significantly improved their internalizing problems (i.e., anxiety, depression, and somatization) as measured by the Behavioral Assessment System for Children. A randomized, controlled intervention study [26] compared primary coping scores on the Response to Stress Questionnaire prior to and after sixth-grade female students completed a mindfulness-based intervention or a control condition for six weeks. This study found no significant interactions between the control or intervention groups and primary coping skills. A separate study found that in a small sample of elementary school youth diagnosed with anxiety or depression, an eight-week-long mindfulness-based intervention did not improve scores of anxiety or depression on the Behavior Assessment System for Children [27]. The need for further longitudinal studies is clear, as it is posited that the effects of yoga interventions only emerge in the long term (~6–7 months post-intervention) [28]. One reason for mixed findings is the lack of fidelity assessments and the difficulty delivering yoga-based practices with fidelity [29]. Attempts to examine the effectiveness of school-based yoga highlight methodological limitations beyond intervention length, such as lack of controls, lack of experimental designs, few replicable interventions, and little fidelity monitoring of intervention curriculums [30, 31].

The present study reports the results of a program evaluation of a universal health and wellness curriculum, Pure Power, designed to teach youth yoga techniques, mindfulness, and healthy emotion regulation. The data presented here draws upon a study with implementation and delivery fidelity data that were previously reported [29] (additional detail in the procedures section). In this report, we sought to provide insights on implementation from an outcomes perspective. Our goal was to identify schools where implementation fidelity was maximized to conclude the intervention was delivered successfully and then evaluate the relationship between exposure to the curriculum and relative improvement on areas of coping, emotion regulation/dysregulation, and achievement when compared to schools that experienced greater barriers and difficulty with implementation. Exposure to the Pure Power curriculum was hypothesized to improve children's coping skills and emotion regulation/dysregulation, as well as improve indices of academics as measured by standardized spelling and math scores.

## Method

The study received IRB approval from Stanford University (Approved 09/23/2014, eProtocol # 31715); the original study protocol (IRB approval documents) are available as supplements. This study was initially planned and received IRB approval as a multifaceted examination of brain function, sleep patterns, and biopsychosocial emotional data among at risk youth with a practical program evaluation component [32] of the wellness curriculum. The study was not originally conceived of as a formal clinical trial, but was later registered as a clinical trial at the request of the PLOS ONE editors (ID: NCT06014970). Data from the research protocol on the fMRI [33] and sleep assessments [34] has been previously published. These studies involved smaller subsamples of the larger study. This paper focuses on the largest groups of participants and on students' self-reported program evaluation outcomes following involvement in the wellness curriculum. Allocation was non-randomized. The intervention model was parallel assignment with no masking of condition (students in the wellness curriculum schools knew

they were receiving the wellness curriculum). The original recruitment plan was 800 children but was based on school level recruitment; based on the schools, a total of 1,176 were recruited with 881 participants from high fidelity schools and matched comparisons analyzed in this study (see details under participants and in the flow diagram).

## Participants

A CONSORT flow diagram is presented in Fig 1. The S2–S7 Tables provide details on sample sizes for each condition and comparison group across each of the assessment points. Authors had no access to information that could identify participants during or after data collection. Data were collected from school districts in the San Francisco Bay Area beginning in the fall of 2014. At study entry, 1,176 children (47.1% female) were recruited, with ages ranging between 7 and 13 ($M$ = 9.19, $SD$ = 1.14 from grades 3 to 5). Data included in current analyses came from eight schools in underserved communities in the Bay Area and utilizes three time points taken roughly one year apart. We analyzed data from students who were in schools that completed the intervention with implementation fidelity (n = 4) and in comparison schools (n = 4). Due to variations in fidelity ratings across intervention schools and demographic characteristics of potential control schools, "matched pairs" were created that pair high-fidelity intervention schools with demographically matched control schools (more details below under design, data analyses, and results sections). Demographics of matched pair schools are presented in S1 Table. The sample of youth was predominantly Hispanic (60.1%), followed by American Indian (4.9%), African American (4.1%), Pacific Islander (3.0%), and non-Hispanic White (1.1%). In total, 988 (91.1%) youth met criteria for free or reduced lunch. Comparisons between intervention and control schools indicated that these groups did not differ on age or gender distribution. Dependent on the amount and combination of assessments completed at each of the three time points (e.g., questionnaires, saliva collections, sleep assessment, brain

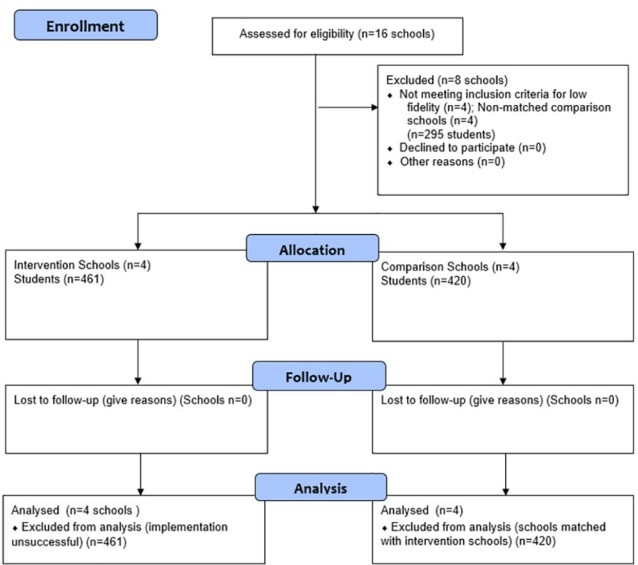

**Fig 1. CONSORT flow diagram.**

scans), students received between $25 and $750 for their participation. Informed consent was obtained from both parents as well as child participants.

## Measures

**The Response to Stress Questionnaire** (RSQ) [35] was used to evaluate youth's coping and emotion regulation strategies. The RSQ is comprised of a checklist of stressors that may be experienced during stressful situations. Following the checklist of stressors, a possible total of 57 questions ask individuals how they respond to stressors they recently experienced. Youth rate how often they cope with a stressor in a certain way on a 4-point Likert scale (ranging from 1, "Not at all," to 4, "A lot"). The RSQ is divided into two major subscales: School-Related Stress Experiences and Stress Reactivity. Stress Reactivity is further divided into two composite scores: involuntary engagement and primary control. Examples of School-Related Stress Experiences include rating how stressful school-related situations have been in the past six months (e.g. "Doing badly on a test or paper," "Having trouble studying," and "Not understanding homework"). The stress reactivity subscale assesses how youth respond to school problems; responses may manifest as rumination (e.g., "I can't stop thinking about what I said or did"), intrusive thoughts (e.g., "Thoughts about school problems just pop into my head"), physiological arousal (e.g., "When I'm dealing with school problems, my heart races"), emotional arousal (e.g., "When I am faced with school problems, I get anxious"), and involuntary action (e.g., "I can't always control what I do"). The Involuntary Engagement composite score is an average of scores on the rumination, intrusive thoughts, physiological arousal, emotional arousal, and involuntary actions subscales. The Primary Control composite score is an average of scores on the emotional expression, emotion regulation, and problem-solving subscales.

**School achievement.** School achievement was assessed with the Wide Range Achievement Test-Revised Spelling and Math scale scores (WRAT-R) [36]. The WRAT-R spelling and math scales were chosen as the academic achievement assessment as it collects a considerable amount of information in a short testing period (ranging from 15 to 45 minutes) and is easy to administer [37]. The spelling subscale was a phonetic test that assessed the degree to which students could encode sounds into written words. The math computation subscale measured students' ability to identify numbers, count, and solve written computational problems.

**Emotional and behavioral dysregulation.** Internalizing symptoms (i.e., anxiety and depression) and externalizing problems (i.e., attention and hyperactivity) were assessed via a self-report version of the Behavior Assessment System for Children, 2nd edition (BASC-2) [38]. The BASC-2 evaluates the self-perception of youth's feelings and behavior. Questions about self-reported anxiety assessed worrying/nervousness (e.g., "I worry most of the day"). High levels of depression indicated youth experienced feelings of sadness, being overwhelmed, or a lack of care (e.g., "I feel like I just don't care anymore"). For attention problems, high scores indicated problematic levels of paying attention (e.g., "I have trouble paying attention to the teacher"), while a high score of hyperactivity suggested problematic levels of activity (e.g., "I have trouble sitting still").

## Procedures

Pure Power is a yoga and mindfulness curriculum developed and delivered between 2014–2018. The curriculum was delivered in K-8 suburban public schools. The school district serves a historically underserved community. The University Institutional Review Board approved this study, which complied with the Helsinki Declaration guidelines. Families were informed of the curriculum and study prior to consenting. Signed assent forms were also obtained from participating students. A consent form written in both English and Spanish was distributed to

families prior to the start of the study. Due to the sample size and minimal risk posed by the study, a waiver of documentation of informed consent was obtained from the Institutional Review Board and implemented with agreement from the participating schools. Researchers explained the study to students prior to starting, and signed assent was obtained from all students who agreed to participate. Students' caregivers were given the opportunity to opt students out of the study by signing and returning an opt-out form to the school.

The manualized Pure Power curriculum (pureedgeinc.org/curriculum) was delivered as an assigned class by yoga instructors to students twice weekly. Full details on Pure Power instructors, including training procedures, are provided in Espil et al. [29]. Pure Power sessions took place in designated school classrooms and were an average of 35 minutes. Children were supplied with yoga mats. Fidelity measurements were taken from the beginning of class, throughout the instruction, and ended when all youth finished the session. Study assessment measures were administered in the classroom setting at the designated assessment time points (baseline pre-intervention, 1-year follow-up, and approximate 2-year follow-up). Study personnel read all RSQ and BASC-2 questions and answer options aloud. Students were instructed to silently circle their answers and not share their answers with other students. Students received no assistance on how to complete WRAT math or spelling problems.

Yoga and mindfulness instruction were delivered in four parts: open breathing, content instruction, posture/movement instruction, and rest. Opening breathing: students were first instructed in a brief breathing exercise. Examples of breathing exercises include "anchor breathing" (i.e., placing hands on the chest or stomach, breathing to feel the body movement with each breath, and focusing the mind) and more active breathing exercises (i.e., audibly inhaling through and exhaling out the nose). Following the opening breathing exercise, students would either transition to rest, a posture or movement task, or content instruction. Content instruction: the curriculum included brief, five-minute content instruction. Themes in content instruction included different "powers," such as the power to tame temper, focus, be calm, and cultivate growth. Modules also included topics about neuroscience. The content delivery was flexible and could occur at any point in the curriculum, either as an independent session segment or between yoga movements or rest. Posture/Movement instruction: the majority (70%) of the session time was allotted to posture and movement instruction. Instructors were given a teaching guide from a manual. The general sequence of posture/movement instruction was as follows: 1) warm-up postures, 2) balancing or standing poses, 3) backbends and floor postures, and 4) closing poses that led to a final resting posture. Rest: during the rest period (4 minutes), students were instructed to lie down on their backs, close their eyes, and concentrate on breathing and sensations (curriculum is available at PureEdgeInc.org in both English and Spanish).

Fidelity Observation Procedure. Details of fidelity training, assessment monitoring, and outcomes are in Espil et al. (2021). Overall, the conclusion was that many aspects of the intervention could be delivered with high reliability and that intervention fidelity varied by identified schools. We address this in the data analytic plan. More information about the structure of fidelity monitoring and analysis is described by Espil et al. [29].

## Design and data analysis plan

As noted above, due to variations in fidelity ratings across intervention schools and demographic characteristics of potential control schools, "matched pairs" were created that pair high-fidelity intervention schools with demographically matched control schools. The pairs were created by authors in the intervention team. When conducting the initial analyses, the data analysis team was initially masked to intervention versus control groups. The outcome

was then presented to the intervention team and then unmasked for writing this report. In total, there were four matched pairs ("Pair A," "Pair B," "Pair C," and "Pair D"), wherein the intervention school in "Pair A" demonstrated the best fidelity, and the intervention school in "Pair D" had relatively lower fidelity.

To determine differences in scores on the RSQ, BASC-2, and spelling and math achievement on the WRAT-R, we conducted 3-way repeated measures factorial ANOVA [2 (group) x 3 (time) x 4 (matched pairs)] for each outcome. In the event of significant interactions, post-hoc analyses were performed to interpret patterns of interactions. All data were analyzed using the Statistical Package for Social Sciences (SPSS) Version 28. With (n = 461) intervention students (n = 420) in matched comparison schools, the study is well powered to identify all repeated measures, between, and between -within interactions. Power analysis indicated sufficient power to detect medium to large effects with power estimates well above .95. Power analysis using G*Power [39, 40] indicated that the power to detect small effects for the most complicated analyses (using effect size $f$ = .1, and desired power = .95, alpha = .050, two tailed) would require 912 denominator degrees of freedom and total n = 464. Power to detect small effects with desired power = .80 (alpha = .050, two tailed) would require 608 denominator degrees of freedom and total n = 312. The above suggests all analyses were power above .80 with most in the above .90-.95 range.

## Results

### Implementation

Upon completion of data collection, it was determined by the implementation team that not all intervention-targeted schools adequately received the intervention protocol and that there was general variability in implementation fidelity across schools. The implementation team reviewed facilitator notes and fidelity data [29] and determined which schools maintained the highest fidelity to the curriculum during implementation. For each school, an overall school-specific fidelity score was calculated by computing the mean fidelity score across all fidelity rating items that were found to have statistically significant reliability. Based on the availability of four control schools with sufficient sample data to conduct matched comparisons, the four intervention schools with the highest overall mean fidelity rating scores were selected for matching and further analysis. Of note, the four intervention schools with the highest overall fidelity scores corresponded with the schools independently identified by the intervention facilitators as having been most successful—with the overall highest quality of instruction and fewest barriers to implementation—during the intervention period.

As noted, the implementation team then matched control schools to these treatment schools based on grade range, school size, percentage of students receiving free or reduced lunch, and percentage of students identified as English Language Learners. The implementation team then sent the data analysis team at ISU a masked data set. The exclusion of problematic cases with invalid or concerning response patterns (as identified by "extreme caution" results on BASC validity scales for the V "nonsensical" Index and Response Pattern Index) did not alter analysis results or significant findings.

**Outcomes.** All missing data were handled analysis-by-analysis. Examination of the distribution of scores indicated non-normal distribution across externalizing variables, with attention displaying a mildly high kurtosis of 2.10 at Time 1. When Mauchly's test of sphericity indicated the assumption of sphericity had been violated and equal variances could not be assumed, the degrees of freedom for within-subjects effects were modified via the Greenhouse-Geisser procedure. When equal variances could not be assumed, post-hoc tests were supplemented with non-parametric alternatives (i.e., Friedman's test). In the case of

differences in significant findings between parametric and non-parametric alternatives, differences are described in the supplement (S-p. 2).

**RSQ coping: Primary control.** ANOVA results for the RSQ primary control composite and subscales are listed in Table 1. Overall, for the primary control composite, there was a significant effect of time [$F$ (1.91, 999.27) = 4.79, $p < .01$], no time x study group interaction, no time x matched pair interaction, but there was a significant effect of time x study group x matched pair [$F$ (5.74, 999.27) = 2.40, $p < .05$]. Post-hoc pairwise comparisons indicated all youth, regardless of the study group, scored lower on Time 1 compared to Time 2 (mean difference = .05, SE = .02, $p < .05$) and Time 3 (mean difference = .07, SE = .02, $p < .001$). This effect is illustrated in Fig 2, and further details are provided in S1 Table in the supplemental information, including n's for all cells and detailed differences.

For the emotion regulation subscale, there was no effect of time, a significant time x study group interaction [$F$ (2, 1026) = 3.15, $p < .05$], no time x matched pair interaction, and no effect of time x study group x matched pair. Post-hoc pairwise comparisons indicated that the control group scored significantly higher than the intervention group at Time 1 (mean difference = .16, SE = .04, $p < .001$) with the intervention group scores increasing to the level of the control group by Time 3. The finding is illustrated in Fig 2 (additional details in S2 Table).

For the emotion expression subscale, there was a significant effect of time [$F$ (1.93, 997.92) = 5.69, $p < .05$], no time x study group interaction, no time x matched pairs interaction, and a significant effect of time x study group x matched pair [$F$ (5.78, 997.92) = 2.47, $p < .05$]. Post-hoc analyses on matched pairs suggested that effects were largest in Pair D. Matched pair effects are detailed in supplemental information (S2 Table), and the overall effect is illustrated in Fig 2. For the problem-solving subscale, there was a significant effect of time [$F$ (1.96, 1011.31) = 5.18, $p < .01$] with scores increasing over time, no time x study group interaction, no time x matched pair interaction, and no effect of time x study group x matched pair.

**Table 1. RSQ primary control and subscales ANOVA summary effects of time, study group, and matched pair.**

| Effect | F | df | p |
|---|---|---|---|
| Primary Control Composite | | | |
| Time | 4.79 | 1.91, 999.27 | < .01 |
| Time x Study Group | .85 | 1.91, 999.27 | .43 |
| Time x Matched Pair | 1.44 | 5.74, 999.27 | .20 |
| Time x Study Group x Matched Pair | 2.40 | 5.74, 999.27 | < .05 |
| Emotion Expression | | | |
| Time | 5.69 | 1.93, 997.92 | < .05 |
| Time x Study Group | 1.89 | 1.93, 997.92 | .15 |
| Time x Matched Pair | 1.26 | 5.78, 997.92 | .28 |
| Time x Study Group x Matched Pair | 2.47 | 5.78, 997.92 | < .05 |
| Emotion Regulation | | | |
| Time | 1.25 | 2, 1026 | .29 |
| Time x Study Group | 3.15 | 2, 1026 | < .05 |
| Time x Matched Pair | 1.88 | 6, 1026 | .08 |
| Time x Study Group x Matched Pair | 1.30 | 6, 1026 | .26 |
| Problem-solving | | | |
| Time | 5.18 | 1.96, 1011.31 | < .01 |
| Time x Study Group | .70 | 1.96, 1011.31 | .50 |
| Time x Matched Pair | 1.16 | 5.89, 1011.31 | .33 |
| Time x Study Group x Matched Pair | 1.09 | 5.89, 1011.31 | .37 |

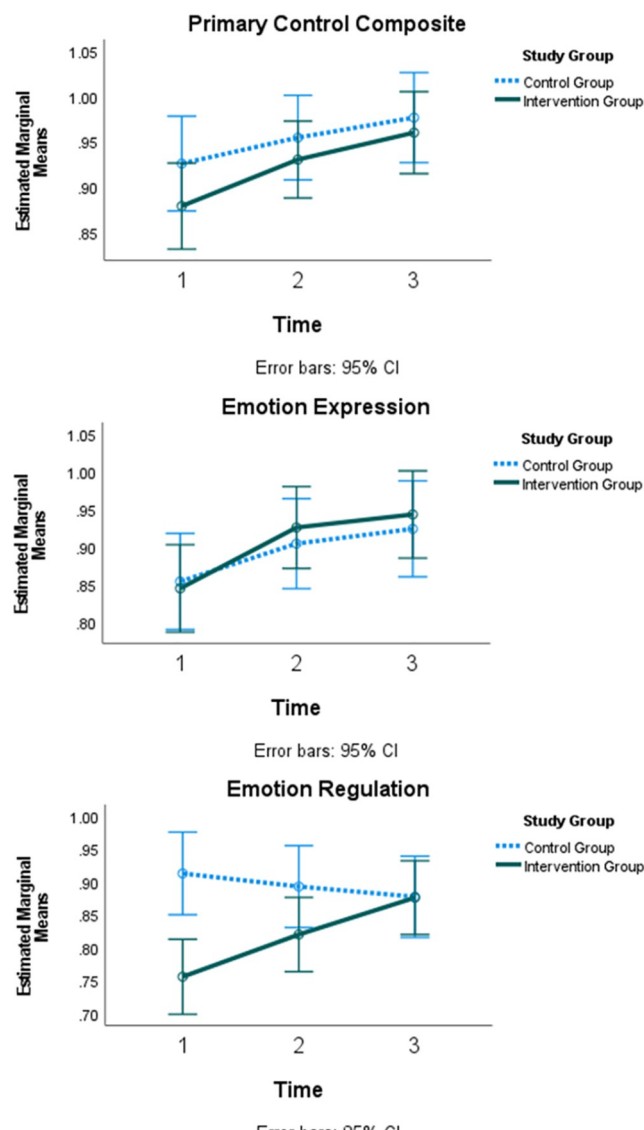

**Fig 2. Differential effect of the intervention on RSQ primary control composite and subscales.**

**WRAT: Academic achievement.** Repeated measures ANOVA results for academic achievement scores are listed in Table 2 and effects are illustrated in Fig 3. For spelling scores, results indicated there was a significant effect of time [$F (1.93, 825.92) = 532.45$, $p < .001$] with scores increasing over time, a significant time x study group interaction [$F (1.93, 825.92) = 15.82$, $p < .001$], a significant time x matched pair interaction [$F (5.78, 825.92) = 2.17$, $p < .05$], and a significant effect of time x study group x matched pair [$F (3.85, 825.92) = 9.80$, $p < .001$]. Post-hoc analyses indicated the control group performed significantly higher (mean difference = 1.03, SE = .51, $p < .05$) than the intervention group at Time 1. However, at Time 2 and Time 3, there were no significant differences in mean scores ($p > .05$). This interaction is plotted in Fig 3. Additionally, the treatment group significantly improved scores from Time 1 to Time 2 and Time 1 to Time 3. Further post-hoc analyses indicated that effects were largest in Pair B (see S3 Table in supplement for details).

**Table 2. ANOVA summary on WRAT spelling and math effects of time, study group, and matched pair.**

| Effect | F | df | p |
|---|---|---|---|
| Spelling Scores | | | |
| Time | 532.45 | 1.93, 825.92 | < .001 |
| Time x Study Group | 15.82 | 1.93, 825.92 | < .001 |
| Time x Matched Pair | 2.17 | 5.78, 825.92 | < .05 |
| Time x Study Group x Matched Pair | 9.80 | 3.85, 825.92 | < .001 |
| Math Scores | | | |
| Time | 246.22 | 1.79, 797.48 | < .001 |
| Time x Study Group | 4.06 | 1.79, 797.48 | < .05 |
| Time x Matched Pair | 2.36 | 5.38, 797.48 | < .05 |
| Time x Study Group x Matched Pair | 3.25 | 3.58, 797.48 | < .05 |

For math scores, results indicated a significant effect of time [$F$ (1.79, 797.48) = 246.22, $p <$ .001] with scores increasing over time, a significant time x study group interaction [$F$ (1.79, 797.48) = 4.06, $p <$ .05], a significant time x matched pair interaction [$F$ (5.38, 797.48) = 2.36, $p <$ .05], and a significant effect of time x study group x matched pair [$F$ (3.58, 797.48) = 3.25, $p <$ .05]. Post-hoc analyses indicated the control group scored significantly higher than the intervention group at Time 1 (mean difference = 1.54, SE = .47, $p <$ .001) and Time 3 (mean difference = 1.34, SE = .56, $p <$ .05). There were no differences in math scores at Time 2. This interaction is plotted in Fig 3. Further post-hoc analyses indicated that schools in Pair A performed higher than other pairs at multiple time points. Full details are provided in S3 Table in supplemental information.

There were no clear patterns of differential effects of the intervention on the BASC subscales nor the RSQ Involuntary Engagement Composite or Subscales. Details of these analyses are in the supplemental analyses text and in the S4–S7 Tables and S1 Checklist.

## Discussion

Yoga and mindfulness instruction may aid in addressing the negative impacts of stress and adversity on emotional and behavioral functioning in adults [14, 19]. Interest in the effectiveness of mindfulness-based interventions such as yoga in primary schools has grown as an avenue for primary prevention. This study adds to the literature which suggests some promise [20–24]. On the positive side, exposure to the Pure Power curriculum was associated with improved children's coping skills, emotion regulation, indices of academics as measured by spelling and math scores. However, consistent with previous literature [22, 23], our results were mixed with a primary take-home finding that implementation (and therefore sustainability) of the program is challenging. Schools demonstrated variability in the fidelity of intervention implementation. For example, unforeseen circumstances had an impact on curriculum delivery, such as cancellation of classes for health-related reasons, variability of support from administration and teachers at individual schools, variability in how schools managed behavioral concerns (e.g., sending students out of class instead of keeping them in class), and the necessity to at times use substitute instructors, such as in the case of illness or injury, of which who may not have had the same experience level and rapport with students as the regular teacher. Program evaluation of these types of programs need to be flexible in fitting into the routine and the unforeseen vicissitudes of the school context [7], such as variability in staff support and facilities, as well as the complex logistics of delivering such programming [29]. To address the sustainability of the program, the Pure Power curriculum has evolved to target

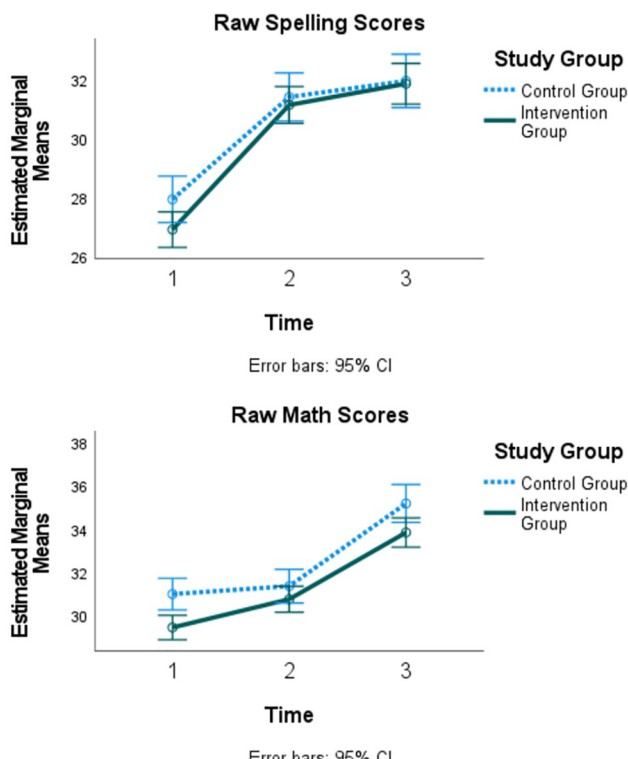

**Fig 3. Differential effect of the intervention on WRAT spelling and math.**

teachers as both targets of the training and facilitators of the intervention (https://pureedgeinc.org/trainings/). Unlike Yoga instructors, teachers are trained in pedagogy in primary schools and possibly better facilitators of the curriculum because of the training they have engaging a room of school children. Further, in research evaluating yoga and mindfulness programs, it will be important to measure and consider the fidelity of program implementation and to incorporate fidelity measures into research design and analysis plans.

The data presented here adds to the literature and builds upon the implementation and delivery fidelity data that were previously reported [29]. Unlike the study in 12–18-year-olds who reported that youth significantly improved their internalizing scores as measured by the BASC [25], we did not find differential effects on the BASC. Still, our findings were similar to a study who did not report improvement scores of anxiety or depression on the BASC [27]. Again, unlike the study that did not find differential effects on the RSQ scales [26], we did find differential effects. Improved spelling could reflect improved network connections since this has been associated with spelling and math [41, 42] and with improved coping/emotion regulation skills [43]; this may be particularly helpful for kids with limited resources, which reflects the population of this study. Such improved emotion regulation may prevent the development of PTSD [8, 44].

While this study adds to the existing knowledge, there are several limitations. The main limitation is the non-randomized design. Thus, causal effects should not be interpreted, but rather, findings should be interpreted similarly to other longitudinal designs examining temporal associations. Although common in effectiveness trials conducting treatment research in community mental health settings [45, 46], loss of data due to dropout at post-intervention and follow-up may have lowered the ability to draw conclusions about treatment

outcomes. Another limitation is that multiple schools were not able to organize sufficient time or space to implement the intervention. This limitation highlights that, while the program may have feasibility, there is a need for additional work on implementation in certain contexts [47]. Similarly, while we observed positive developmental trajectories in some outcomes in the intervention group, not everyone improved on all outcomes at follow-up. Finally, we did not measure academic outcomes with grades; we used spelling and math test scores as rated by school administrators, counselors, or teachers using the WRAT-R. An important future step will be to analyze academic outcomes in other areas and collect data from other stakeholders.

In conclusion, challenges to implementing yoga and mindfulness-based programming in real-life settings are vital to identify. This study provided some real-world evidence for the effectiveness of a universal health and wellness curriculum on emotion regulation and positive academic outcomes. Given the results, training school staff to effectively and consistently deliver the intervention may foster implementation. Future research should test the effectiveness of who delivers the intervention; for example, teacher-delivered groups vs. other wellness personnel. Ideally, training school staff to deliver the intervention could foster implementation by utilizing individuals trained in teaching grade-school classrooms. Testing the effectiveness of teacher-delivered groups in contrast to external facilitators is an important future research direction.

## Supporting information

**S1 File.**
(PDF)

**S1 Checklist.** *PLOS ONE* **clinical studies checklist.**
(DOCX)

**S1 Table. Demographic distributions by study group and paired schools (% at Time 1).**
(DOCX)

**S2 Table. Means, standard deviations, and effect size estimates for RSQ primary control scores.**
(DOCX)

**S3 Table. Means, standard deviations, and effect size estimates on WRAT scores.**
(DOCX)

**S4 Table. Summary of ANOVAs.** Effects of time, study group, and matched pair on RSQ Involuntary Engagement and subscales.
(DOCX)

**S5 Table. Summary of ANOVAs.** Effects of time, study group, and matched pair on BASC scale scores.
(DOCX)

**S6 Table. Means, standard deviations, and effect size on BASC scores.**
(DOCX)

**S7 Table. Means, standard deviations, and effect size estimates on RSQ involuntary engagement scores.**
(DOCX)

## Author Contributions

**Conceptualization:** Travis Bradley, Ryan Matlow, Carl F. Weems, Victor G. Carrion.

**Data curation:** Bethany H. McCurdy, Travis Bradley, Ryan Matlow, Flint M. Espil.

**Formal analysis:** Bethany H. McCurdy, Carl F. Weems.

**Funding acquisition:** Victor G. Carrion.

**Investigation:** Victor G. Carrion.

**Project administration:** Victor G. Carrion.

**Supervision:** Carl F. Weems, Victor G. Carrion.

**Writing – original draft:** Bethany H. McCurdy.

**Writing – review & editing:** Bethany H. McCurdy, Travis Bradley, Ryan Matlow, John P. Rettger, Flint M. Espil, Carl F. Weems, Victor G. Carrion.

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
