## [Decision Letter · Decision Letter 0]

25 Oct 2023

PONE-D-23-21703Program Evaluation of a School-Based Mental Health and Wellness Curriculum Featuring Yoga and MindfulnessPLOS ONE

Dear Dr. McCurdy,

Thank you for submitting your manuscript to PLOS ONE. After careful consideration, we feel that it has merit but does not fully meet PLOS ONE’s publication criteria as it currently stands. Therefore, we invite you to submit a revised version of the manuscript that addresses the points raised during the review process.

Please note that we have only been able to secure a single reviewer to assess your manuscript. We are issuing a decision on your manuscript at this point to prevent further delays in the evaluation of your manuscript. Please be aware that the editor who handles your revised manuscript might find it necessary to invite additional reviewers to assess this work once the revised manuscript is submitted. However, we will aim to proceed on the basis of this single review if possible. 

We look forward to receiving your revised manuscript.

Kind regards,

Avanti Dey, PhD

Staff Editor

PLOS ONE

2. Please remove your figures from within your manuscript file, leaving only the individual TIFF/EPS image files, uploaded separately. These will be automatically included in the reviewers’ PDF.

Reviewers' comments:

Reviewer's Responses to Questions

**Comments to the Author**

1. Is the manuscript technically sound, and do the data support the conclusions?

Reviewer #1: Yes

2. Has the statistical analysis been performed appropriately and rigorously? 

Reviewer #1: Yes

3. Have the authors made all data underlying the findings in their manuscript fully available?

Reviewer #1: Yes

4. Is the manuscript presented in an intelligible fashion and written in standard English?

Reviewer #1: Yes

5. Review Comments to the Author

Reviewer #1: Overall, this manuscript is clear, well-organized and written and offers important information about the challenges of program implementation. The authors are thorough in their reporting of the study justification, design and procedures. I have detailed specific comments below. I wonder if there could be more commentary on the challenges associated with program implementation since this could shed more light on study findings and be informative for readers.

Specific Comments:

• Overall, introduction is very nicely written and supported with relevant research. The background moves nicely from the problem to coping to mindfulness to yoga and the importance of fidelity and study design.

• P.10, bottom of first full paragraph – is nonattachment the only pathway through which mindfulness is theorized or shown to aid in emotion regulation?

• Bottom of p.10 – can the age ranges of the samples being discussed here be specified?

• Top of p.11: “internalizing scores” may require some explanation.

• In the discussion, I wonder if the authors could provide more contextual information about the challenges with implementation – factors that could have affected the results and more suggestions for future research. This seems to be the most valuable part of this study. I also wonder if there needs to be more discussion on why so few findings were significant as expected.

6. PLOS authors have the option to publish the peer review history of their article (what does this mean?). If published, this will include your full peer review and any attached files.

Reviewer #1: No

---

## [Author Response · Author response to Decision Letter 0]

29 Nov 2023

Reviewer Response Letter

November 21, 2023

Avanti Dey 

Staff Editor

Dear Dr. Dey,

We extend our gratitude for inviting a second revision of our paper titled, “Program Evaluation of a School-Based Mental Health and Wellness Curriculum Featuring Yoga and Mindfulness” and for the thoughtful comments we received from the current reviewer. The review we have received has improved the quality of our manuscript. We have addressed each comment in the revised manuscript or in this response letter. In the following, we detail how we addressed each point in the order in which they appeared in your letter. We include the reviewer’s comments followed by our response to the comment with page numbers as appropriate.

COMMENTS FOR THE AUTHOR:

Reviewer #1: Overall, this manuscript is clear, well-organized and written and offers important information about the challenges of program implementation. The authors are thorough in their reporting of the study justification, design and procedures. I have detailed specific comments below. I wonder if there could be more commentary on the challenges associated with program implementation since this could shed more light on study findings and be informative for readers.

Specific Comments:

• Overall, introduction is very nicely written and supported with relevant research. The background moves nicely from the problem to coping to mindfulness to yoga and the importance of fidelity and study design.

We thank the reviewer very much for their constructive and encouraging comments on our introduction. With their feedback, we will ensure that this positive momentum is maintained throughout the entire manuscript.

• P.10, bottom of first full paragraph – is nonattachment the only pathway through which mindfulness is theorized or shown to aid in emotion regulation?

We thank the reviewer for pointing out this oversight. We have included more examples (with a new citation) of pathways through which mindfulness is thought to enhance solution-focused strategies and emotion regulation, such as through promoting goal orientation, problem disengagement, and mobilizing resources to focus on acceptance and awareness [page 10 of pdf, page 4 of manuscript]. This sentence now reads: “Effectively, mindfulness is thought to enhance the ability to engage in solution-focused strategies and emotion regulation through promoting goal orientation, problem disengagement, and mobilizing resources to focus on acceptance and awareness while minimizing avoidance-oriented coping by fostering an attitude of non-attachment towards stressful life experiences [15, 16].”

• Bottom of p.10 – can the age ranges of the samples being discussed here be specified?

Thanks for this suggestion. We double-checked and the specific ages or ranges were not reported for all papers; however, the grade ranges of youth in studies cited here are available and are now referenced in the text [page 10-11 of pdf, page 4-5 of manuscript].This sentence now reads: “Preliminary evidence shows some promise, as youth in second grade to twelfth grade who engage in yoga to promote mindfulness show improved coping skills, increased socio-emotional competence and prosocial skills, decreased alcohol use, and teacher-rated improvements in academic performance, attention span, and ability to deal with stress and feelings of anxiety [20-24].”

• Top of p.11: “internalizing scores” may require some explanation.

We have edited this sentence so it now reads: “For example, a multi-year longitudinal study [25] measuring the effectiveness of a mindfulness-based stress reduction intervention in 12–18-year-olds found that youth significantly improved their internalizing problems (i.e., anxiety, depression, and somatization) as measured by the Behavioral Assessment System for Children.” [page 11 of pdf, page 5 of manuscript]

• In the discussion, I wonder if the authors could provide more contextual information about the challenges with implementation – factors that could have affected the results and more suggestions for future research. This seems to be the most valuable part of this study. I also wonder if there needs to be more discussion on why so few findings were significant as expected.

We thank the reviewer for this suggestion, as the logistics of delivering this program in school settings, sustainability, and challenges when conducting a program evaluation are critical points made within this manuscript that address a strong need for consideration in future programming and evaluation. We believe the challenges in implementation attributed to mixed findings. To elaborate on these points, we have added the following text [page 22 of pdf, page 16 of manuscript]:

“Schools demonstrated variability in the fidelity of intervention implementation. For example, unforeseen circumstances had an impact on curriculum delivery, such as cancellation of classes for health-related reasons, variability of support from administration and teachers at individual schools, variability in how schools managed behavioral concerns (e.g., sending students out of class instead of keeping them in class), and the necessity to at times use substitute instructors, such as in the case of illness or injury, of which who may not have had the same experience level and rapport with students as the regular teacher. Program evaluation of these types of programs need to be flexible in fitting into the routine and the unforeseen vicissitudes of the school context [7], such as variability in staff support and facilities, as well as the complex logistics of delivering such programming [29].”

Note that we included a duplicate reference in the manuscript and the duplicate is now removed.

In sum, we have considered each suggestion and have addressed each point. We once again

thank you for your time and close read of our manuscript. We look forward to hearing from you.

Sincerely, 

The Authors

---

## [Decision Letter · Decision Letter 1]

20 Feb 2024

PONE-D-23-21703R1Program Evaluation of a School-Based Mental Health and Wellness Curriculum Featuring Yoga and MindfulnessPLOS ONE

Dear Dr. McCurdy, Thank you for submitting your revised manuscript to PLOS ONE. Your manuscript has been reviewed by 2 reviewers, one of whom still has some outstanding minor concerns as outlined in the comments below. Could you please address their comments?

We look forward to receiving your revised manuscript.

Kind regards,

Annesha Sil, PhD

Associate Editor, PLOS ONE

Journal Requirements:

Reviewers' comments:

Reviewer's Responses to Questions

**Comments to the Author**

1. If the authors have adequately addressed your comments raised in a previous round of review and you feel that this manuscript is now acceptable for publication, you may indicate that here to bypass the “Comments to the Author” section, enter your conflict of interest statement in the “Confidential to Editor” section, and submit your "Accept" recommendation.

Reviewer #1: All comments have been addressed

Reviewer #2: (No Response)

2. Is the manuscript technically sound, and do the data support the conclusions?

Reviewer #1: Yes

Reviewer #2: No

3. Has the statistical analysis been performed appropriately and rigorously? 

Reviewer #1: Yes

Reviewer #2: No

4. Have the authors made all data underlying the findings in their manuscript fully available?

Reviewer #1: Yes

Reviewer #2: No

5. Is the manuscript presented in an intelligible fashion and written in standard English?

Reviewer #1: Yes

Reviewer #2: No

6. Review Comments to the Author

Reviewer #1: (No Response)

Reviewer #2: Although this is a Revision, I think the statistical analysis has been presented half-heartedly, and would require further work for improvement. I state those below.

1. Abstract: In the Results, statements of direction ("relative improvement on measures of emotion....") should be accompanied by a p-value.

2. Sample size/power: It was quite strange to find no sample size/power justification, considering the matched design, using the primary response variable. Please also mention the statistical test, 1-, or 2-sided, 5% level of significance, and desired effect size. Note, the manuscript is submitted as a "Clinical Trial", where, sample size/power statements are essential. Create a separate subsection

3. Statistical analysis: The proposed 3-way repeated measures factorial ANOVA is based wholesomely on Gaussian assumptions. There is no mention of the (nonparametric) alternatives to consider under violations to those assumptions, such as the Friedman's test. During analysis, Gaussianty checks should be performed.

7. PLOS authors have the option to publish the peer review history of their article (what does this mean?). If published, this will include your full peer review and any attached files.

Reviewer #1: No

Reviewer #2: No

---

## [Author Response · Author response to Decision Letter 1]

22 Feb 2024

February 21, 2024 Response to Reviewers

Annesha Sil

Associate Editor

RE: PONE-D-23-21703R1

Program Evaluation of a School-Based Mental Health and Wellness Curriculum Featuring Yoga and Mindfulness

PLOS ONE

Dear Dr. Sil,

Thanks so much for your decision requesting additional minor revisions. One reviewer indicated that we had addressed all concerns and one reviewer outlined outstanding minor concerns, which we have addressed in our response below (in italics). We thank all reviewers again for the excellent feedback to improve the paper.

Reviewer 1: Had no additional comments and confirmed our responsiveness to the previous comments. We again thank this reviewer for the helpful comments to strengthen our paper.

Reviewer 2:

Although this is a Revision, I think the statistical analysis has been presented half-heartedly, and would require further work for improvement. I state those below.

We first thank the reviewer for the suggestions for improvement below and address each point. We do disagree about our analysis presentation as being “half-hearted”. The supplemental information contains a plethora of statistical and descriptive data and some information on testing assumptions was actually in the previous draft (see below).

1. Abstract: In the Results, statements of direction ("relative improvement on measures of emotion....") should be accompanied by a p-value.

We wrote this paper using guidance supplied by the American Psychological Association (APA) and Journal of the American Medical Association (JAMA). APA style does not require p-values in abstracts. Indeed, many writing guides suggest to not include: “Note: Generally speaking, specific values and data (such as percentages, standard errors, p-values, etc.) should not be included in the abstract. Rather, this part of the abstract should provide readers with an overview.” https://www.cwauthors.com/article/what-to-include-and-exclude-in-an-abstract

Moreover, most importantly, emphasizing a p-value in an abstract is inconsistent with best practices vis JAMA: see Halabi, S., & Day, S. (2019). Improved Reporting in Abstracts When Uncertainty Is Inevitable. JAMA Network Open, 2(12), e1917543-e1917543.

We feel that reporting a p-value in the abstract distracts from the more complicated picture of findings we obtained. However, we remain open to the editor’s guidance on this matter.

2. Sample size/power: It was quite strange to find no sample size/power justification, considering the matched design, using the primary response variable. Please also mention the statistical test, 1-, or 2-sided, 5% level of significance, and desired effect size. Note, the manuscript is submitted as a "Clinical Trial", where, sample size/power statements are essential. Create a separate subsection

We think it’s important to point out -- as is noted in the paper on page 6 -- “The study was not originally conceived of as a formal clinical trial, but was later registered as a clinical trial at the request of the PLOS ONE editors (ID: NCT06014970).” We would also note that we have provided extensive detail on sample size in the manuscript or supplement (e.g., detailed cell sizes), each analysis has degrees of freedom reported, and estimates of effect size. We concur we did not include our power analyses in the draft. To improve clarity, we have added this information to our data analysis section (p. 6-7): 

“With (n= 461) intervention students (n= 420) in matched comparison schools, the study is well powered to identify all repeated measures, between, and between -within interactions. Power analysis indicated sufficient power to detect medium to large effects with power estimates well above .95. Power analysis using g power indicated that the power to detect small effects for the most complicated analyses (using effect size f = .1, and desired power = .95, alpha = .050, two tailed) would require 912 denominator degrees of freedom and total n = 464. Power to detect small effects with desired power = .80 (again alpha = .050, two tailed) would require 608 denominator degrees of freedom and total n = 312. The above suggests all analyses were power above .80 with most in the above .90-.95 range.”

The following citations were added: 

Faul, F., Erdfelder, E., Buchner, A., & Lang, A.-G. (2009). Statistical power analyzes using G*Power 3.1: Tests for correlation and regression analyses. Behavior Research Methods , 41 , 1149-1160.

Faul, F., Erdfelder, E., Lang, A.-G., & Buchner, A. (2007). G*Power 3: A flexible statistical power analysis program for the social, behavioral, and biomedical sciences. Behavior Research Methods , 39 , 175-191

3. Statistical analysis: The proposed 3-way repeated measures factorial ANOVA is based wholesomely on Gaussian assumptions. There is no mention of the (nonparametric) alternatives to consider under violations to those assumptions, such as the Friedman's test. During analysis, Gaussianty checks should be performed.

To our knowledge there is no non-parametric alternative to a 3-way repeated measures factorial ANOVA as there is for t-tests and ANOVA’s. However, we tested assumptions and reported on page 13 in the submitted draft “When Mauchly's test of sphericity indicated the assumption of sphericity had been violated and equal variances could not be assumed, the degrees of freedom for within-subjects effects were modified via the Greenhouse-Geisser procedure.”

To further clarify, we also specify that data were screened for non-normal distributions violations of skew and kurtosis. It was found that the BASC variable Attention displayed non-normal distributions (now more specifically described on p. 13); notably, the kurtosis value was mildly high (a value of 2.10). We have now added the additional clarification that post hoc tests were supplemented with tests that did not assume equal variances, specifying our use of the Friedman test. In the case of differences in significant findings between parametric and non-parametric alternatives, differences were described. (p. 13, Supplement p. 2). 

We thank you and the reviewers again for their thorough review and suggestions to improve the quality of the manuscript. We look forward to hearing from you.

Sincerely,

The Authors

---

## [Decision Letter · Decision Letter 2]

11 Mar 2024

Program Evaluation of a School-Based Mental Health and Wellness Curriculum Featuring Yoga and Mindfulness

PONE-D-23-21703R2

Dear Dr. Bethany McCurdy,

We’re pleased to inform you that your manuscript has been judged scientifically suitable for publication and will be formally accepted for publication once it meets all outstanding technical requirements.

Kind regards,

Sandra Julia Diller

Academic Editor

PLOS ONE

Reviewers' comments: Reviewer #2: All comments have been addressed

---

## [Editor Report · Acceptance letter]

26 Mar 2024

PONE-D-23-21703R2 

PLOS ONE

Dear Dr. McCurdy, 

I'm pleased to inform you that your manuscript has been deemed suitable for publication in PLOS ONE. Congratulations! Your manuscript is now being handed over to our production team.

Kind regards, 

on behalf of

Dr. Sandra Julia Diller 

Academic Editor

PLOS ONE